# Star Image Prediction and Restoration under Dynamic Conditions

**DOI:** 10.3390/s19081890

**Published:** 2019-04-20

**Authors:** Di Liu, Xiyuan Chen, Xiao Liu, Chunfeng Shi

**Affiliations:** 1Key Laboratory of Micro-Inertial Instrument and Advanced Navigation Technology, Ministry of Education, School of Instrument Science and Engineering, Southeast University, Nanjing 210096, China; sdili_liudi@163.com (D.L.); sidescan@126.com (X.L.); shchfeng@163.com (C.S.); 2School of Electrical Engineering and Automation, Qilu University of Technology, Jinan 250353, China

**Keywords:** star image prediction, star sensor, Richardson-Lucy algorithm, neural network

## Abstract

The star sensor is widely used in attitude control systems of spacecraft for attitude measurement. However, under high dynamic conditions, frame loss and smearing of the star image may appear and result in decreased accuracy or even failure of the star centroid extraction and attitude determination. To improve the performance of the star sensor under dynamic conditions, a gyroscope-assisted star image prediction method and an improved Richardson-Lucy (RL) algorithm based on the ensemble back-propagation neural network (EBPNN) are proposed. First, for the frame loss problem of the star sensor, considering the distortion of the star sensor lens, a prediction model of the star spot position is obtained by the angular rates of the gyroscope. Second, to restore the smearing star image, the point spread function (PSF) is calculated by the angular velocity of the gyroscope. Then, we use the EBPNN to predict the number of iterations required by the RL algorithm to complete the star image deblurring. Finally, simulation experiments are performed to verify the effectiveness and real-time of the proposed algorithm.

## 1. Introduction

Along with the development of navigation technology, the requirement for a spacecraft attitude measurement is becoming higher and higher [1,2]. In general, star sensors and gyroscopes are often used in spacecraft to measure the attitude information. The star sensor is supposed to be the most accurate attitude-measuring device in stable conditions [3]. However, under dynamic conditions, frame loss and blurring of the star image may occur, which leads to decreased accuracy or even failure of the star centroid extraction and attitude determination. Therefore, only by solving the frame loss and blurring problem of the star image, can the star sensor maintain good performance under dynamic conditions. Because gyroscopes have a relatively high measurement accuracy and excellent dynamic performance in a short period, using the gyroscope to assist in improving the dynamic performance of the star sensor has become a hot topic [4,5,6,7,8,9].

In the process of spacecraft motion, due to the influence of external interference and the limitation of the star sensor, the star sensor is prone to frame loss, which can result in a lack of coherence in the process of moving image tracking and even loss of key motion features. Therefore, how to eliminate the frame loss error has become a research hotspot in the field of image processing. Currently, the primary methods for eliminating frame loss error includes the frame loss error elimination based on the support vector machine (SVM) [10,11], frame loss error elimination based on iterative error compensation [12,13] and frame loss error elimination based on adaptive minimum error threshold segmentation [14]. These methods eliminate the interference noise in the image and compensate the frame loss error, but still cannot avoid the frame loss. To overcome the shortcomings of the above methods, a method for eliminating the frame loss by using a motion image-tracking model is presented in [15], since the frame loss of the star image is mainly affected by the exposure time and readout time of the star sensor [2]. Therefore, in [16,17,18], parallel processing is used to overlap exposure time and readout time to reduce the frame loss of the star image. In [19,20], the authors used image intensifiers to increase the sensitivity of the image sensor, thereby reducing the occurrence of the frame loss in the star sensors. In [21], Wang et al. proposed using field programmable gate arrays (FPGAs) to improve the processing ability of the star sensor to reduce the readout time. Yu et al. [22] proposed a method to reduce the occurrence of the frame loss by using an intensified star sensor. Although FPGAs and image intensifiers can assist the star sensor in reducing the occurrence of the frame loss, the additional FPGA and image intensifier lead to an increase in the weight and power consumption of the star sensor and limit its application in micro-spacecraft.

The motion blur of the star image is another important reason that affects the dynamic performance of the star sensor. To improve the dynamic performance of the star sensor, many scholars have done a lot of research in the field of image processing, especially on the star image deblurring algorithms [23]. According to whether the point spread function (PSF) is known or not, the deblurring methods can be classified into two typical forms: Blind image deblurring (BID) with unknown PSF, and non-blind image deblurring (NBID) with known PSF [24]. Mathematically, the process of NBID is an inverse problem, and an NBID algorithm has a good real-time performance. Currently, most BID algorithms perform blur kernel estimation and image deblurring simultaneously, and recursively to approach the sharp image [25,26,27,28,29,30]. Therefore, BID methods have poor real-time performance. Because star sensors are widely used in spacecraft, the real-time requirements are high. Therefore, we intend to study an NBID algorithm for star image deblurring.

Two problems should be solved in the process of restoring the blurred star image. One is how to determine the blur kernel, and the other is to choose which deblurring method to use. The gyroscope can be used to measure the angular rates of the carrier and is easy to integrate, and the blur kernel parameters (blur angle and blur length) can be calculated according to the angular rate information output by the gyroscope. In this paper, a gyroscope is used to assist in the calculation of blur kernels. For the star image deblurring, there are two commonly used NBID algorithms. One is the Wiener filter [31,32]. Quan et al. [31] proposed a Wiener filter based on the optimal window technique for recovering the blurred star image. Ma et al. [32] proposed an improved one-dimensional Wiener filtering method for star image deblurring. Although the two methods are better in real time, they also amplify the noise in the image. The other is the Richardson–Lucy (RL) algorithm, which can effectively suppress the noise in the deblurred star image [33,34]. However, the iterative convergence criterion is not given in the RL algorithm, and the optimal number of iterations needs to be obtained through constant-trying with large time-consumption. If the amount of blurred star image to be processed is enormous, this is a disadvantage that cannot be ignored.

In this paper, to solve the shortcomings of the above methods and further improve the performance of the star sensor under highly dynamic conditions, we propose an improved gyroscope-assisted star image prediction method and RL non-blind deblurring algorithm. In the star image prediction method, considering the second-order distortion of the star sensor lens, a prediction model between the angular rates of the gyroscope and the position of the star spot is established. For the improved RL algorithm, first, we analyze the point spread function (PSF) model of the star sensor under different motion conditions, and then the ensemble back-propagation neural network (EBPNN) prediction model based on the improved bagging method is constructed to predict the number of termination iterations required by the conventional RL algorithm, which is used to overcome the disadvantage of traditional RL algorithm that needs to set the number of iterations manually.

The rest of this paper is organized as follows. In Section 2, we introduce the star image prediction model in the case of the frame loss of the star image. The improved RL algorithm is described in Section 3. In Section 4, simulation results are shown to demonstrate the effectiveness of our method. Finally, we give a conclusion in Section 5.

## 2. Prediction Model of the Star Image

The star sensor is a vision sensor that can be used to measure the attitude of a spacecraft [35]. To obtain the high-precision attitude information of the spacecraft, we must ensure that the image sensor of the star sensor can output the star image continuously. Due to the highly dynamic motion of the spacecraft, frame loss of the star image often occurs. Therefore, it is especially important to ensure that the star sensor can output the high-precision attitude information under the condition of the frame loss of the star image. In this section, we will show how to predict the position of the star spot based on the angular rates of the gyroscope in the presence of distortion of the star sensor lens. In Figure 1, the star sensor obtains the direction vector of the navigation star in the celestial inertial coordinate system by observing the stars on the celestial sphere. At time t, the attitude matrix of the star sensor in the celestial coordinate system is A(t), the star sensor can detect the direction vector $v_{i}$ of the navigation star in the celestial coordinate system, and its image vector can be represented as $W_{i}$ in the star sensor coordinate system. The image coordinate of the principal point of the lens of the star sensor is $\left(x_{0},y_{0}\right)$, the coordinates of the navigation star $S_{i}$ on the image plane is $\left(x_{i},y_{i}\right)$. Since the optical lens of the star sensor mainly has a second-order radial distortion, the ideal image coordinate $\left(x_{i}^{\prime },y_{i}^{\prime }\right)$ of the navigation star $S_{i}$ can be expressed as,
(1)$$\left\{ \begin{array}{l} x_{i}^{\prime }-x_{0}=(x_{i}-x_{0})(1+k_{x}^{\prime }\cdot r^{2}),\\ y_{i}^{\prime }-y_{0}=(y_{i}-y_{0})(1+k_{y}^{\prime }\cdot r^{2}), \end{array}\right.$$
where, $r=\sqrt{{\left(x_{i}-x_{0}\right)}^{2}+{\left(y_{i}-y_{0}\right)}^{2}}$, $k_{x}^{\prime }$ and $k_{y}^{\prime }$ represent the second-order radial distortion coefficients in the X and Y directions, respectively.

Assuming that the focal length of the star sensor is f, the direction vector $W_{i}$ can be given by
(2)$$W_{i}=\frac{1}{\sqrt{{\left[\left(x_{i}-x_{0})(1+k_{x}^{\prime }\cdot r^{2}\right)\right]}^{2}+{\left[\left(y_{i}-y_{0})(1+k_{y}^{\prime }\cdot r^{2}\right)\right]}^{2}+f^{2}}}\left[\begin{array}{c} \left(x_{i}-x_{0})(1+k_{x}^{\prime }\cdot r^{2}\right)\\ \left(y_{i}-y_{0})(1+k_{y}^{\prime }\cdot r^{2}\right)\\ -f \end{array}\right].$$

According to the attitude matrix A(t) of the star sensor, the relationship between the vectors $W_{i}$ and $v_{i}$ can be obtained,
(3)$$W_{i}=A(t)\cdot v_{i},$$
where, the attitude matrix A(t) can be solved by the N vector method, Trial method, Quest method, Q-method and Least square method [36]. In this paper, we use the angular velocity information of the gyroscope to calculate the attitude matrix A(t).

In Figure 2, $O_{S}XYZ$ represents the star sensor coordinate system, $O_{C}uv$ represents the image plane coordinate system, the projection point of the principal point $O_{S}$ of the lens of the star sensor on the image plane is $O_{C}$, $O_{C}O_{S}$ is consistent with the principal optical axis of the star sensor lens and its length is equal to the focal length f. $w_{x}$, $w_{y}$ and $w_{z}$ represent the three-axis angular rates of the star sensor at instant t, which can be measured by the gyroscope. P denotes the position of the navigation star on the star image at instant t, $O_{C}P$ denotes the direction vector of the navigation star under the coordinate system of the star sensor, and the star spot P shifts to $P^{\prime }$ at instant t+∆t. According to Equation (3), the direction vectors $\vec{O_{S}P}$ and $\vec{O_{S}P^{\prime }}$ can be expressed as,
(4)$$\left\{ \begin{array}{l} \vec{O_{S}P}=W_{i}(t)=A(t)\cdot v_{i},\\ \vec{O_{S}P^{\prime }}=W_{i}(t+∆t)=A(t+∆t)\cdot v_{i}, \end{array}\right.$$
where, $A(t+∆t)=A_{t}^{t+∆t}\cdot A(t)$, A(t+∆t) denotes the attitude matrix at instant t+∆t.
(5)$$\begin{array}{ll} A_{t}^{t+∆t} & =I-(w(t)\times )\cdot ∆t=I-\left[\begin{array}{ccc} 0 & -w_{z}(t) & w_{y}(t)\\ w_{z}(t) & 0 & -w_{x}(t)\\ -w_{y}(t) & w_{x}(t) & 0 \end{array}\right]\cdot ∆t\\ & =\;\left[\begin{array}{ccc} 1 & w_{z}(t)\cdot ∆t & -w_{y}(t)\cdot ∆t\\ -w_{z}(t)\cdot ∆t & 1 & w_{x}(t)\cdot ∆t\\ w_{y}(t)\cdot ∆t & -w_{x}(t)\cdot ∆t & 1 \end{array}\right], \end{array}$$
where (w(t)×) represents the cross-product matrix of the star sensor angular rates vector w(t).

According to Equations (4) and (5), the relationship between $W_{i}(t)$ and $W_{i}(t+∆t)$ can be obtained,
(6)$$W_{i}(t+∆t)=A_{t}^{t+∆t}\cdot W_{i}(t),$$
where, we can calculate $W_{i}$ through the star image. According to Equations (1) and (6), we can obtain the position prediction model as follows,
(7)$$\left\{ \begin{array}{l} x_{i}^{\prime }(t+∆t)=\frac{(x_{i}(t)-x_{0})(1+k_{x}^{\prime }\cdot r^{2})+x_{0}+((y_{i}(t)-y_{0})(1+k_{y}^{\prime }\cdot r^{2})+y_{0})\cdot w_{z}(t)\cdot ∆t+f\cdot w_{y}(t)\cdot ∆t}{\left(-((x_{i}(t)-x_{0})(1+k_{x}^{\prime }\cdot r^{2})+x_{0})\cdot w_{y}(t)\cdot ∆t+((y_{i}(t)-y_{0})(1+k_{y}^{\prime }\cdot r^{2})+y_{0})\cdot w_{x}(t)\cdot ∆t\right)/f+1}\\ y_{i}^{\prime }(t+∆t)=\frac{((y_{i}(t)-y_{0})(1+k_{y}^{\prime }\cdot r^{2})+y_{0})-((x_{i}(t)-x_{0})(1+k_{x}^{\prime }\cdot r^{2})+x_{0})\cdot w_{z}(t)\cdot ∆t-f\cdot w_{x}(t)\cdot ∆t}{\left(-((x_{i}(t)-x_{0})(1+k_{x}^{\prime }\cdot r^{2})+x_{0})\cdot w_{y}(t)\cdot ∆t+((y_{i}(t)-y_{0})(1+k_{y}^{\prime }\cdot r^{2})+y_{0})\cdot w_{x}(t)\cdot ∆t\right)/f+1} \end{array}\right..$$

## 3. Improved Star Image Deblurring Algorithm

Generally, establishing a PSF under a specific motion is the key to star image recovery. In this section, first, we analyze the PSF model of the blurred star image caused by the rotation of the star sensor around the optical axis and non-optical axis and calculate the PSF in the corresponding motion condition through the angular velocity information of the gyroscope. Then, we introduce an improved RL algorithm to recover the blurred star image.

### 3.1. Motion Blur Model of the Star Image

To better recover the blurred star image, the primary task is to obtain the PSF. Therefore, it is necessary to analyze the mechanism of the star image blurs. The star sensor is a navigation device that acquires the attitude by utilizing star observations. Because the star sensor needs to photograph the sky with a dark background, in order to increase the number of navigation stars in the star image, it needs to increase the exposure time appropriately. If the star sensor has a wide range of motion during the exposure time, the same star will be imaged at different locations on the star image, which will result in blurring of the star image. Mathematically, the model of star image blurring can be written as,
(8)g(x,y)=f(x,y)⊗h(x,y)+n(x,y),
where f(x,y), g(x,y), and h(x,y) denote the sharp star image, the blurred star image, and the PSF, respectively; ⊗ represents two-dimensional convolution operator, and n(x,y) denotes the image noise.

Due to the different motion types of the star, sensors produce different PSFs, so PSF is important for describing the model of the blurred star image. Since the distance from the navigation star to the earth is much larger than the distance from the star sensor to the earth, the linear motion has less effect on the star image blur, and this effect can be ignored. Therefore, we mainly analyze the model of the blurred star image generated by the angular motion.

In Figure 3a, the star image blur caused by the angular motion is shown. Since the exposure time of the star sensor is short, the angular velocity of the star sensor can be considered to be constant during the exposure time. Moreover, the star sensor coordinate system is coincident with the body-fixed frame. In Figure 3b, the model of the blurred star image generated by the star sensor rotating around the X-axis is shown, the initial angle between the starlight direction and the principal optical axis of the star sensor is α, and the projection of the navigation star is P in the star image. When the star sensor rotates clockwise around the X-axis at an angular velocity $w_{x}$, and during the exposure time ∆t, the rotational angle is $∆\alpha =w_{x}∆t$, and the star spot moves from P to $P^{\prime }$ in the image plane. The geometric relationship between P and $P^{\prime }$ is,
(9)$$L_{PP^{\prime }}=f\cdot \left[\tan(\alpha +∆\alpha )-\tan\alpha \right]/d_{ccd},$$
where $L_{PP^{\prime }}$ represents the distance from P to $P^{\prime }$ quantized by pixels, $d_{ccd}$ denotes the pixel size, and f is the focal length of the star sensor.

As a result of the short exposure time of the star sensor, ∆α is quite small, the first order Taylor-expansion for tan(α+∆α) can be obtained.
(10)$$\begin{array}{ll} \tan(\alpha +∆\alpha ) & \approx \tan\alpha +(\tan\alpha {)}^{\prime }\cdot ∆\alpha \\ & =\tan\alpha +\left(\frac{\sin^{2}\alpha +\cos^{2}\alpha }{\cos^{2}\alpha }\right)\cdot ∆\alpha \\ & =\tan\alpha +(\tan^{2}\alpha +1)\cdot ∆\alpha . \end{array}$$

Substituting Equation (10) into (9), we have
(11)$$L_{PP^{\prime }}=f\cdot (\tan^{2}\alpha +1)\cdot ∆\alpha /d_{ccd}.$$

In general, the rotational motion characteristics of the star sensor in the $O_{S}X$ and $O_{S}Y$ directions are the same. As shown in Figure 3c, during the exposure time ∆t, the star sensor rotates clockwise around the Y-axis at an angular velocity $w_{y}$, the rotational angle is $∆\alpha^{\prime }=w_{y}∆t$, the star spot shifts along the u axis in the image plane, and its translation vector can be obtained.
(12)$$L_{PP^{\prime }}=f\cdot (\tan^{2}\alpha +1)\cdot ∆\alpha^{\prime }/d_{ccd}.$$

When the star sensor rotates around the X-axis and Y-axis with angular rates $w_{x}$ and $w_{y}$, respectively, and after the exposure time ∆t, the rotation angle of the star sensor is $∆\alpha^{\prime \prime }=w_{xy}\cdot ∆t=\sqrt{w_{x}^{2}+w_{y}^{2}}∆t$, and the translation vector of the star spot is
(13)$$L_{PP^{\prime }}=f\cdot (\tan^{2}\alpha +1)\cdot ∆\alpha^{\prime \prime }/d_{ccd}.$$

In general, when the star sensor rotates around the cross bore-sight direction ($O_{S}X$ and *O_S_Y* directions), the blur kernel angle θ of the star image can be given by
(14)$$\theta =\arctan\left[\frac{\tan(\alpha +∆\alpha )-\tan\alpha }{\tan(\alpha +∆\alpha^{\prime })-\tan\alpha }\right].$$

Then, the PSF of the blurred star image is expressed as [37,38],
(15)$$h_{1}(x,y)=\left\{ \begin{array}{l} 1/L_{PP^{\prime }},\;if\;y/x=\sin\left|\theta \right|/\cos\left|\theta \right|,0\leq x\leq L_{PP^{\prime }}\cdot \cos\left|\theta \right|\\ 0,\;otherwise \end{array}\right..$$

In Figure 3d, the star sensor rotates clockwise around the Z-axis at an angular rate $w_{z}$, point P(u,v) does a circular motion with $O_{C}$ as the center and $r=\sqrt{u^{2}+v^{2}}$ as the radius. The rotation angle of the star sensor is $∆\alpha^{\prime \prime \prime }=w_{z}\cdot ∆t$ during the exposure time ∆t. Since the exposure time of the star sensor is short, the arc length $PP^{\prime }$ can be approximated as the chord length $∆PP^{\prime }$. Inspired by reference [39], the motion of the star spot can be regarded as a uniform linear motion on the focal plane. The displacement of the star spot in the direction of the X- and Y-axis can be expressed as,
(16)$$\left\{ \begin{array}{l} ∆PP_{u}^{\prime }\approx -v\cdot w_{z}\cdot ∆t,\\ ∆PP_{v}^{\prime }\approx u\cdot w_{z}\cdot ∆t. \end{array}\right.$$

The star image blur kernel angle θ and the $∆PP^{\prime }$ are given by
(17)$$\theta =\arctan(∆PP_{u}^{\prime }/∆PP_{v}^{\prime }),$$
(18)$$\begin{array}{ll} ∆PP^{\prime }= & \sqrt{∆PP_{u}^{\prime 2}+∆PP_{v}^{\prime 2}}\\ & =\sqrt{v^{2}\cdot w_{z}^{2}\cdot ∆t^{2}+u^{2}\cdot w_{z}^{2}\cdot ∆t^{2}}\\ & =\left|w_{z}\right|\cdot ∆t\cdot r. \end{array}$$

According to the geometric relation in Figure 3d,
(19)$$\tan\alpha =r\cdot d_{ccd}/f.$$

Substituting Equation (19) into Equation (18), Equation (18) can be rewritten as
(20)$$∆PP^{\prime }=\left|w_{z}\right|\cdot ∆t\cdot f\cdot \tan\alpha /d_{ccd}.$$

Therefore, when the star sensor rotates around the Z-axis, the PSF of the blurred star image is expressed as,
(21)$$h_{2}(x,y)=\left\{ \begin{array}{l} 1/∆PP^{\prime },if\;y/x=\sin\left|\theta \right|/\cos\left|\theta \right|,0\leq x<∆PP^{\prime }\cdot \cos\left|\theta \right|\\ 0,otherwise \end{array}\right..$$

In summary, according to Equations (15) and (21), the model of the multiple-blurred star image is given by
(22)$$g(x,y)=f(x,y)⊗h_{1}(x,y)⊗h_{2}(x,y)+n(x,y),$$
where the $h_{1}(x,y)$ and $h_{2}(x,y)$ need to be calculated based on the angular velocity $w_{x}$, $w_{y}$ and $w_{z}$ of the star sensor. In this paper, we use a gyroscope to provide the angular velocity $\left[w_{bx}\;w_{by}\;w_{bz}\right]$ of the spacecraft. Therefore, the angular velocity $\left[w_{x}\;w_{y}\;w_{z}\right]$ of the star sensor is expressed as,
(23)$${\left[w_{x},w_{y},w_{z}\right]}^{T}=C_{b}^{s}{\left[w_{bx},w_{by},w_{bz}\right]}^{T},$$
where $C_{b}^{s}$ denotes the rotation matrix from the body coordinate system to the star sensor coordinate system. Because the star sensor is fixed on the spacecraft, $C_{b}^{s}$ can be calibrated in advance.

After obtaining the PSF, the NBID algorithm is used to recover the blurred star image.

### 3.2. Richardson-Lucy (RL) Algorithm

The NBID algorithm includes both linear and nonlinear algorithms. The most common linear NBID algorithms include the inverse filtering algorithm, Wiener filtering algorithm, and least squares algorithm [3]. Compared with the linear NBID algorithm, nonlinear NBID algorithm has a better effect in suppressing noise and preserving image edge information. Currently, the RL algorithm [40] is the most widely used non-linear iterative restoration algorithm. The RL algorithm is a blurred image deconvolution algorithm that extends from the maximum a posteriori probability estimate. This method assumes that the noise in the image follows a Poisson distribution, and the likelihood probability of the image is
(24)$$p(g/f)=\prod_{x,y}\frac{({\left(f(x,y)⊗h(x,y)\right)}^{g(x,y)}e^{-(f(x,y)⊗h(x,y))}}{g(x,y)!},$$
where, (x,y) denotes the pixel coordinate, g(x,y) represents the blurred image, h(x,y) denotes the PSF, and ⊗ denotes the two-dimensional convolution operator.

To get the maximum likelihood solution of the sharp image f(x,y), we minimize the energy function.
(25)$$E(f)=\sum_{x,y}\{(f(x,y)⊗h(x,y))-g(x,y)\cdot \log(f(x,y)⊗h(x,y))(x,y)\}.$$

By deriving the E(f) and normalizing the blur kernel h(x,y), the RL algorithm iteratively updates the image by
(26)$$f^{n+1}(x,y)=[\frac{g(x,y)}{f^{n}(x,y)⊗h(x,y)}⊗h(-x,-y)]f^{n}(x,y),$$
where n represents the iteration number.

The RL algorithm has two important properties [40]: Non-negativity and energy preserving. It constrains the non-negative of estimated values of the sharp image and preserves the total energy of the image in the iteration so that the RL algorithm has excellent performance in the star image deblurring. However, the iterative convergence criterion is not given in the RL algorithm, and the optimal number of iterations need to be obtained through constant-trying with large time-consumption. This shortcoming of the RL algorithm cannot be ignored if we are dealing with a large number of the blurred star image. Therefore, it is necessary to study an improved RL algorithm which automatically sets the number of iterations.

### 3.3. Improved RL Algorithm

To overcome the shortcomings of the RL algorithm, we propose an improved RL algorithm, and the flow diagram is shown in Figure 4. First, we set the parameters of the star sensor including the field of view, focal length, star magnitude limit, resolution of the star image, etc. We use these parameters to simulate a large number of sharp star image and the corresponding blurred star image. Second, according to the angular rates of the gyroscope output, we calculate the PSF of each blurred star image and use the RL algorithm to deblur the star image and record the optimal number of iterations used. The optimal number of iterations and the sum of the Magnitude of Fourier Coefficients (SUMFC) of the PSF of the blurred star image are used for the training of the ensemble back-propagation neural network (EBPNN) [41]. After the training is completed, the optimal iteration number prediction model of the RL algorithm can be obtained. Finally, when the navigation system is used, the PSF of the blurred star image is obtained according to the angular velocity of the gyroscope, and the SUMFC of the PSF is used as the input of the prediction model. The star image is deblurred according to the number of iterations required by the RL algorithm of the prediction model output. Especially, when predicting the number of iterations, the ensemble back-propagation neural network (EBPNN) prediction model based on the improved bagging method uses the SUMFC of the PSF of the blurred star image as the input.

We use different PSFs to blur the sharp star image (Figure 5a). The relationship between the SUMFC of PSFs and the corresponding number of iterations required by the RL algorithm is shown in Figure 6. We can see that there is an obvious non-linear relationship between them, which prompts us to use EBPNN to predict the optimal number of iterations of the RL algorithm.

The performance of a single back-propagation (BP) neural network is limited. It takes a long time to learn, and its objective function is easy to fall into a local minimum. Therefore, we use the integration strategy based on the improved bagging method to integrate the single neural network. The bagging method [42] is based on the re-sampling and self-help technology. The self-help learning sample set $D_{i}(i=1,2,\ldots )$ is retrieved from the original training set D, the size of each self-learning sample set is equivalent to the original training set, and each self-learning sample trains a single BP neural network. The bagging method increases the diversity of neural network by re-selecting the training set, thereby improving the generalization ability and prediction accuracy of the EBPNN.

In order to further improve the prediction accuracy of the ensemble neural network, we introduce a just-in-time learning algorithm to optimize the sample sets $D_{i}(i=1,2,\ldots )$ obtained by the bagging method. Suppose two input samples $x_{i}$ and $x_{q}$, where $x_{q}$ is the currently acquired input sample and $x_{i}$ is a training sample in $D_{i}(i=1,2,\ldots )$. The distance and angle between them can be calculated by the following equation.
(27)$$\left\{ \begin{array}{l} d(x_{i},x_{q})=\sqrt{{\left\|x_{i}-x_{q}\right\|}^{2}},\\ \theta (x_{i},x_{q})=\arccos\frac{x_{i}^{T}x_{q}}{{\left\|x_{i}\right\|}^{2}{\left\|x_{q}\right\|}^{2}}. \end{array}\right.$$

The similarity between $x_{i}$ and $x_{q}$ is
(28)$$S(x_{i},x_{q})=\alpha e^{-d(x_{i},x_{q})}+(1-\alpha )\cos(\theta (x_{i},x_{q})),$$
where, α is the weighting factor, the larger the $S(x_{i},x_{q})$ value, the higher the similarity between $x_{i}$ and $x_{q}$.

We select the k-group data closest to the currently acquired one sample $x_{q}$ from the training sample set $D_{i}(i=1,2,\ldots )$ and arrange the new sample set in descending order.
(29)$$\left\{ \begin{array}{l} D_{i}^{\prime }=\left\{\left(x_{1,i},y_{1,i}),(x_{2,i},y_{2,i}),\cdots ,(x_{k,i},y_{k,i}\right)\right\},i=1,2,\ldots ,\\ S(x_{1},x_{q})>S(x_{2},x_{q})>\cdots >S(x_{k},x_{q}), \end{array}\right.$$
where $y_{k,i}$ denotes the expected output value corresponding to $x_{k,i}$ in training sets $D_{i}(i=1,2,\ldots )$.

Therefore, the local modeling problem is transformed into an optimization problem.
(30)$$J(\delta )=\underset{\delta }{\min}\sum_{i=1}^{k}\left(y_{i}-\hat{y}(\delta ,x_{i}\right){)}^{2}\cdot S(x_{i},x_{q}).$$

Minimize J(δ) to obtain the model parameter δ at the current moment, and then obtain its local model:(31)$$y_{q}=\hat{y}(\delta ,x_{q}).$$

In particular, we find that the computational complexity of the EBPNN model increases with the increase of the number of BPNN models, but the prediction accuracy of the EBPNN model does not always increase with it, sometimes it even decreases. Therefore, after considering the computational complexity and prediction accuracy of the EBPNN model, we decide to use three sub-BP neural network models to construct the EBPNN model. As shown in Figure 7, three BP neural networks are trained by different sample sets $D_{i}^{\prime }(i=1,2,3)$, and the integrated prediction model is obtained by aggregating the three BP neural networks. When the EBPNN is used for prediction, we use the weighted method to integrate the output of each neural network and take the integrated result as the output of EBPNN. In the process of integrating the output of each BP neural network, first, we calculate the average training errors $e_{i}(i=1,2,3)$ of three sub-models on their respective training sample set. Then, we construct a weighting vector w of 1×n dimensions, the value of n is the same as the number of sub-BP neural network models, so n=3, $w_{i}=1/e_{i}$, (i=1,2,3). Finally, we calculate the prediction results of three sub-models for the input data $x_{q}$ by Equation (31) and form a 1×3-dimensional output vector $y_{q}^{\prime }$. The final prediction result of the EBPNN is expressed as,
(32)$$y=\frac{w\cdot y_{q}^{\prime }{}^{T}}{\sum_{i=1}^{3}w_{i}}.$$

To verify the effectiveness of the EBPNN prediction model, we analyzed the accuracy of the iteration times estimated by the model. In the training stage of the EBPNN model, each BP neural network adopts a three-layer structure. The nodes of the input layer, hidden layer, and output layer are set to 1, 10 and 1, respectively. The sigmoid function is used as the activation function. The original training set D contains 1708 samples. In Figure 8, we show the number of iterations predicted by the EBPNN model and compare it with the optimal number of iterations. We can see that the number of iterations estimated by the EBPNN almost coincides with the optimal number of iterations, and the error between them is small. Therefore, the performance of the EBPNN prediction model can meet our requirements.

After EBPNN predicts the number of iterations, we use the improved RL algorithm to obtain the sharp star image, and then we can accurately estimate the attitude information by the star image segmentation, star extraction, star identification, star matching, and other operations [43].

## 4. Simulation Results and Analysis

In order to prove the effectiveness of the star image prediction method and the improved RL algorithm in the highly dynamic environment, we compare and analyze the prediction accuracy of the star spot, and the accuracy of the attitude estimation before and after the star image deblurring in the following section.

### 4.1. Star Image Prediction Experiment

In this section, to validate the star image prediction method, we need to simulate the star image acquired by the star sensor at a different time. In the process of star image simulation, we determine the position of the navigation star in the star image based on the bore-sight direction of the star sensor and the right ascension and declination of the navigation star. Since the star sensor is fixed on the spacecraft, it can obtain different star images as the spacecraft moves. We assume that the exposure time of the star sensor is 0.01 s, the field of view is 20°×20°, the image sensor size is 865 pixels ×
865 pixels, the pixel size is 20 μm, the focal length is 49 mm, and select the stars brighter than 3m in Yale Bright Star Catalogue as the guide star catalog. We use these parameters and the spacecraft trajectory to simulate the images at different times and use them as the ground truth of the star image. According to the above parameters, the resolution of the star image we simulated is 865×865. To speed up the processing of the star image, we intercept the 512×512 size as the star image to be processed. The trajectory of the spacecraft we simulated is shown in Figure 9. And 1500 frames of consecutive star images are simulated, the first and the 1500th frame star image are shown in Figure 10.

To validate the star image prediction method, we predict the star image based on the first frame star image and the angular velocity of the gyroscope, and compare it with the ground truth of the star image. Figure 11a,b show the ground truth of the 1500th frame star image and the 1500th frame star image predicted by the proposed algorithm. To more intuitively demonstrate the accuracy of the prediction algorithm, in Table 1, the centroid coordinates of the star spot in the real and predicted 1500th frame star image is shown, where (x,y) represents the centroid coordinate of the star spot in the real star image, $\left(x^{\prime },y^{\prime }\right)$ is the centroid coordinate of the predicted star spot. ∆x and ∆y represent the difference of the horizontal and vertical coordinates between the true star spot and the predicted star spot, respectively. As seen from Table 1, the maximum error of the horizontal and vertical coordinates of the star spot predicted by our method within 15 s is 0.89 and 0.50 pixels, respectively.

To further analyze the prediction algorithm, according to the first frame star image shown in Figure 10a, we successively predicted the position of stars in 1500 star images, and analyze the mean value of the estimation error of the star spot position in each predicted star image. As shown in Figure 12, the mean value of the coordinate errors of the predicted star spot increases with the increase of the estimated number of frames, but the mean errors could stay in a small range. Therefore, in the case of the short-term frame loss, the proposed method can achieve an accurate prediction of the star spot.

### 4.2. Experiments on Star Image Deblurring

In this section, we present some examples to validate the proposed gyro-assisted improved RL algorithm. First, we analyze the blurring of the star image when the star sensor rotates around the X-axis, the Y-axis, the Z-axis, the X- and Y-axis, and the three axes simultaneously. Then, we add the Gaussian white noise with zero mean and variance 0.01 to the blurred star images. Finally, the blurred star image is deblurred by our proposed algorithm, and we compare the deblurred star image with the original sharp star image. Figure 13 shows the magnified original star image, blurred star images caused by the star sensor rotate around the X- and Y-axis, (*w_x_* = 10°/*s*, $w_{y}=10{}^{\circ}/s$), deblurred star image, and the gray distribution of the star spot in them. As can be seen from Figure 13, the gray value of the star spot in the blurred star image decreases significantly, and after deblurring the star image, the smearing phenomenon is obviously suppressed, the gray value and the gray distribution of star spot are closer to those in original star image.

The star sensor is an attitude measurement device. To more intuitively reflect the deblurring performance of the proposed algorithm, we compare the attitude information of the spacecraft estimated by the star image before and after deblurring. The star image observed by the star sensor at a certain time is shown in Figure 14. First, we perform an angular motion blurring on the observed star image, then we use the proposed algorithm and the automatic iterative RL algorithm to deblur the star image, and compare the attitude information estimated by the deblurred images. The automatic iterative RL algorithm calculates the mean square error (MSE) of the currently restored image by automatically increasing the number of iterations, and compares it with the MSE of the image restored by the last iteration. If the MSE of the currently restored image is higher than the last iteration recovery result, the last iteration number is considered to be the optimal number of iterations, and the restored image is the optimal restoration result. The attitude estimation results are shown in Table 2, Table 3, Table 4, Table 5 and Table 6, and the “Fail” indicates that the attitude information of the spacecraft cannot be estimated by the star image because the degree of blurring of the star image is too high.

From Table 2, Table 3, Table 4, Table 5 and Table 6, it can be seen that the attitude estimation failed when the angular velocity of the star sensor rotating around the X-axis, the Y-axis, the Z-axis, the X-and the Y-axis, and the three axes exceeds $w_{x}=25{}^{\circ}/s$, $w_{y}=30{}^{\circ}/s$, $w_{z}=25{}^{\circ}/s$, w=[20,20,0]°/s and w=[15,15,15]°/s, respectively. After the blurred star image is restored by the proposed algorithm and the automatic iterative RL algorithm, the maximum angular velocity of the attitude can be estimated to be expanded to $w_{x}=75{}^{\circ}/s$, $w_{y}=75{}^{\circ}/s$, $w_{z}=80{}^{\circ}/s$, w=[75,75,0]°/s, and w=[55,55,55]°/s, respectively, these two methods have a similar performance, and the attitude errors are kept in a small range. This is because with the increase of the angular velocity of the star sensor, the blur extent of the star image gets bigger, and the gray value of the star spot decreases significantly. When the gray value of a blurred star is lower than the threshold for star image segmentation and the blurred star can hardly be detected. However, after the restoration of the blurred star image, the gray value of the star spot is improved, and the gray distribution of the star spot is closer to the true distribution so that the star spot can also be extracted under high dynamic conditions. Finally, the attitude of the spacecraft can be estimated by these star spots.

To verify the real-time performance of the proposed algorithm in the case of Gaussian noise, we use the proposed algorithm and the automatic iterative RL algorithm to restore the blurred star image caused by the star sensor rotating around the Z-axis and compare the time consumed by the two methods. As shown in Figure 15, the real-time performance of the proposed algorithm is significantly better than the iterative RL algorithm. This is mainly because the proposed algorithm can use the ensemble neural network based on the improved bagging method to quickly predict the number of iterations required by the RL algorithm, while the iterative RL algorithm requires a step-by-step iteration to optimize the number of iteration steps required.

Second, in the case where the blurred star image is contaminated by Poisson noise, we present the deblurring performance of the proposed method and compare it with the automatic iterative RL algorithm. Figure 16 shows the magnified original star image, blurred star images caused by star sensor rotate around the X- and Y-axis, ($w_{x}=w_{y}=10{}^{\circ}/s$), deblurred star image, and the gray distribution of the star spot in the case of Poisson noise. Combined with Table 7, Table 8, Table 9, Table 10 and Table 11, we can see that in the case of Poisson noise, the attitude estimation failed when the angular velocity of the star sensor rotating around the X-axis, the Y-axis, the Z-axis, the X-and the Y-axis, and the three axes exceeds $w_{x}=40{}^{\circ}/s$, $w_{y}=35{}^{\circ}/s$, $w_{z}=35{}^{\circ}/s$, w=[30,30,0]°/s and w=[15,15,15]°/s, respectively. After the blurred star image is restored by the proposed algorithm and the automatic iterative RL algorithm, the maximum angular velocity of the attitude can be estimated to be expanded to $w_{x}=160{}^{\circ}/s$, $w_{y}=160{}^{\circ}/s$, $w_{z}=170{}^{\circ}/s$, w=[120,120,0]°/s, and w=[80,80,80]°/s, respectively, these two methods have a similar performance, and the attitude errors are kept in a small range. Figure 17 shows the real-time performance of the proposed algorithm and the iterative RL algorithm when dealing with the blurred star image caused by the star sensor rotating around the Z-axis, and the result shows that the real-time performance of our algorithm is better than the iterative RL algorithm when the degree of the blurred star image is large.

In summary, the proposed method and the iterative RL algorithm significantly improve the dynamic performance of the star sensor and have similar performance. However, the real-time performance of our algorithm is better than the iterative RL algorithm, especially in the case of Gaussian white noise.

## 5. Conclusions

In this paper, we improve the dynamic performance of the star sensor by using the star image prediction method and the star image deblurring method. Taking into account the distortion of the star sensor lens, we use the information provided by the star sensor and the gyroscope to establish a star spot prediction model. Also, for the blurred star image problem, we proposed an improved Richardson-Lucy (RL) algorithm based on the EBPNN.

Experimental results demonstrate that the proposed methods are effective in improving the dynamic performance of the star sensor. The maximum error of the star image prediction algorithm is 0.89 pixels in 15 s and the attitude errors calculated from the star image restored by the improved RL algorithm can be kept in a small range. Compared with the iterative RL algorithm, the improved RL algorithm proposed in this paper has better real-time performance.

## Figures and Tables

**Figure 1 sensors-19-01890-f001:**
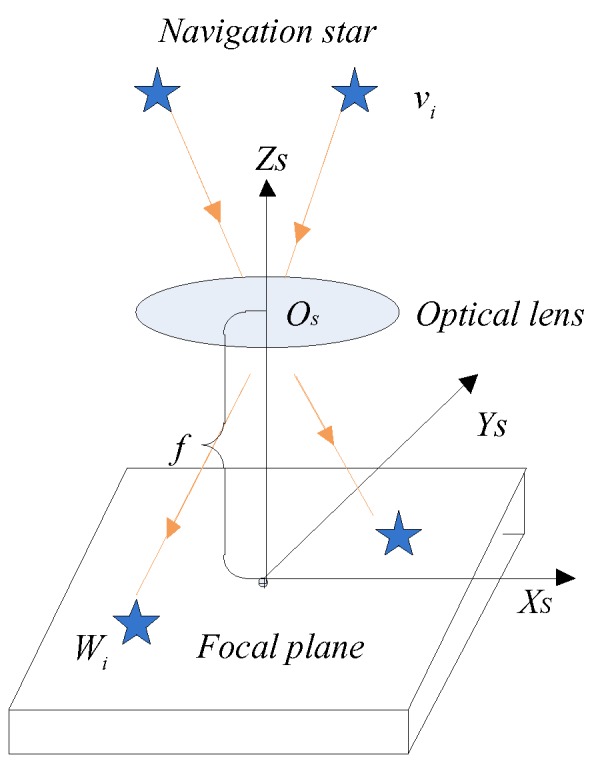
Star image model of the star sensor.

**Figure 2 sensors-19-01890-f002:**
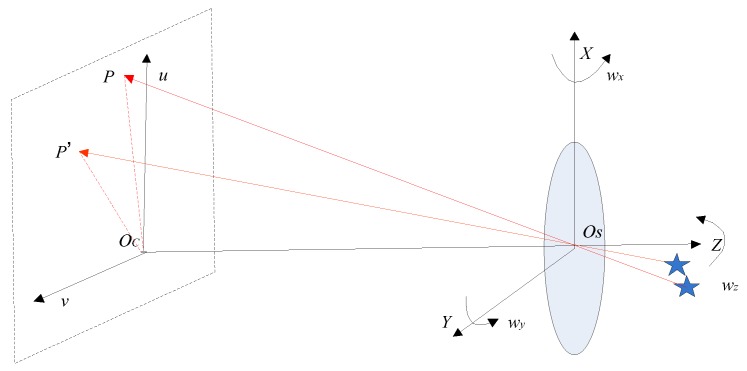
Prediction model of the star spot.

**Figure 3 sensors-19-01890-f003:**
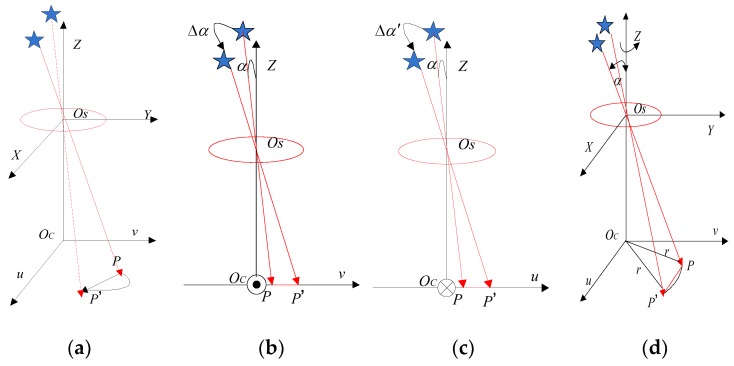
Motion blur star image model. (**a**) Blurred star image generated by the angular motion; (**b**) blurred star image generated by the rotation of the star sensor around the X-axis; (**c**) blurred star image generated by the rotation of the star sensor around the Y-axis; (**d**) blurred star image generated by the rotation of the star sensor around the Z-axis.

**Figure 4 sensors-19-01890-f004:**
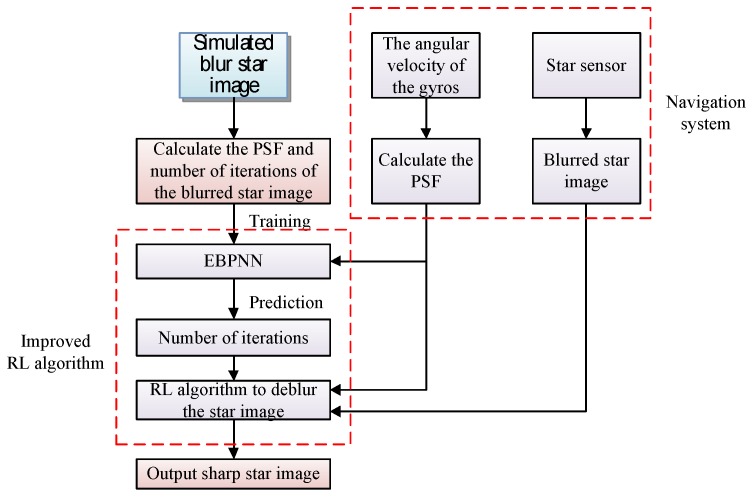
Flow diagram of the improved Richardson-Lucy (RL) algorithm for star image deblurring.

**Figure 5 sensors-19-01890-f005:**
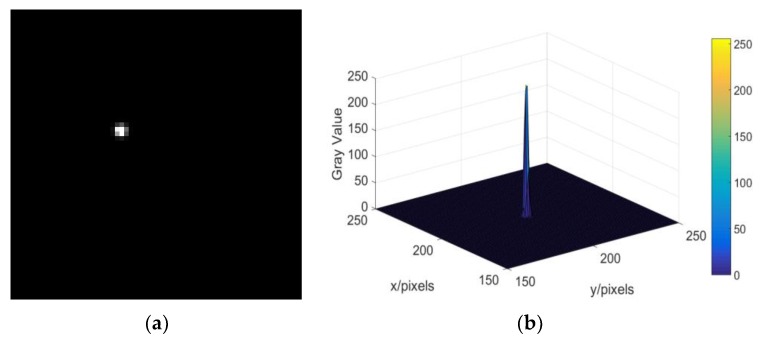
Original sharp star image and its gray distribution. (**a**) Original sharp star image; (**b**) gray distribution of star spot.

**Figure 6 sensors-19-01890-f006:**
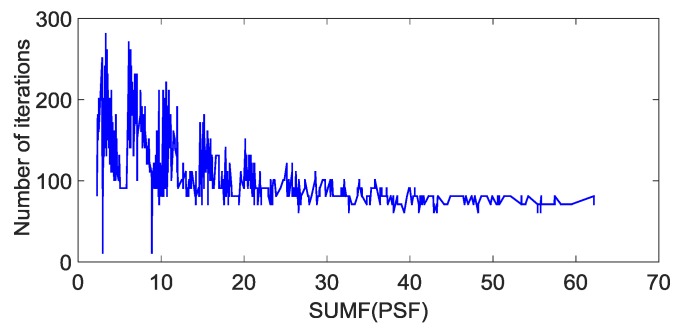
The relationship between the magnitude of Fourier coefficients (SUMFC) of the point spread function (PSF) and the corresponding optimal number of iterations.

**Figure 7 sensors-19-01890-f007:**
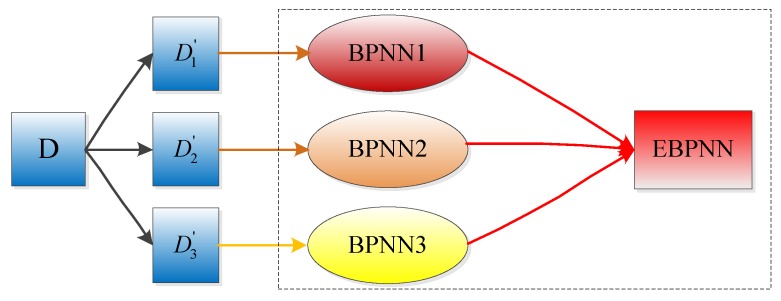
The ensemble back-propagation neural network.

**Figure 8 sensors-19-01890-f008:**
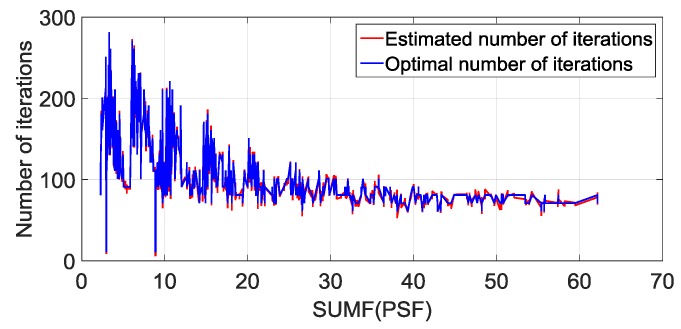
Comparison between the optimal number of iterations and the estimated number of iterations.

**Figure 9 sensors-19-01890-f009:**
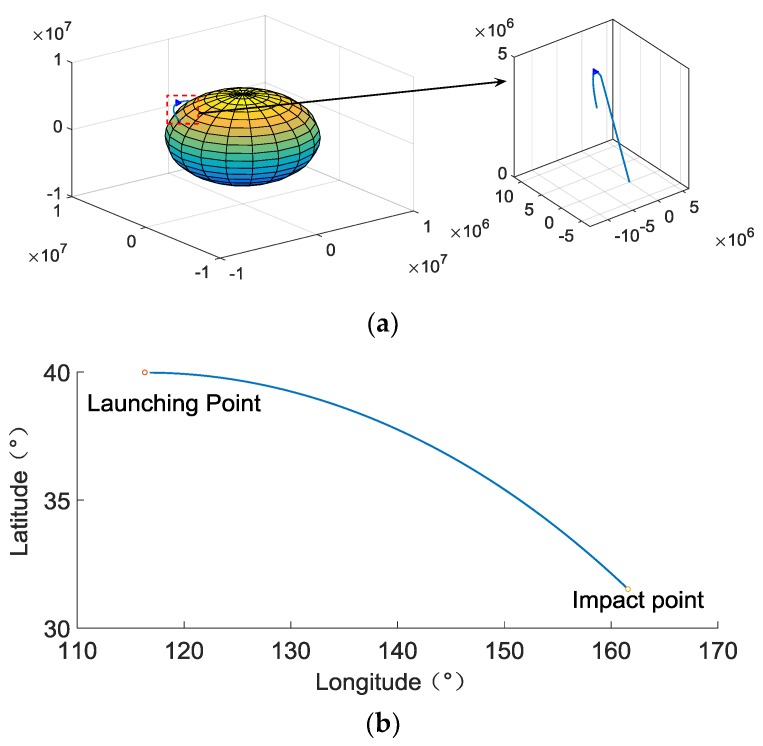
The spacecraft trajectory. (**a**) Three-dimensional trajectory of spacecraft; (**b**) projection of the spacecraft trajectory on the surface of the Earth.

**Figure 10 sensors-19-01890-f010:**
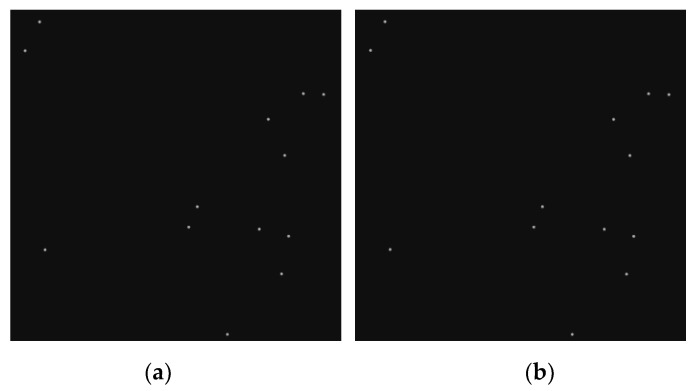
Star image simulation result. (**a**) The first frame star image; (**b**) the 1500th frame star image.

**Figure 11 sensors-19-01890-f011:**
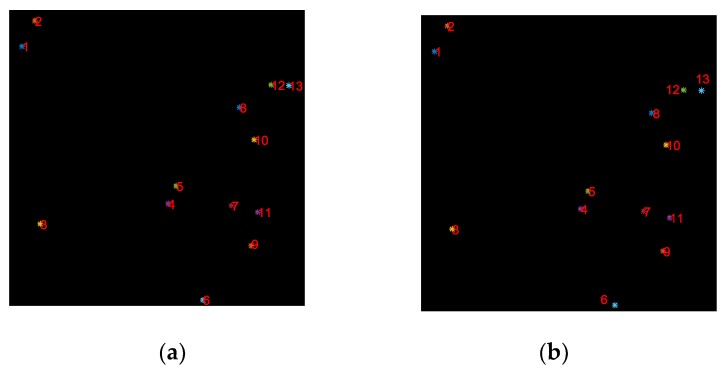
True star image versus predicted star image. (**a**) The true value of the 1500th frame star image; (**b**) the 1500th frame star image predicted by the proposed algorithm.

**Figure 12 sensors-19-01890-f012:**
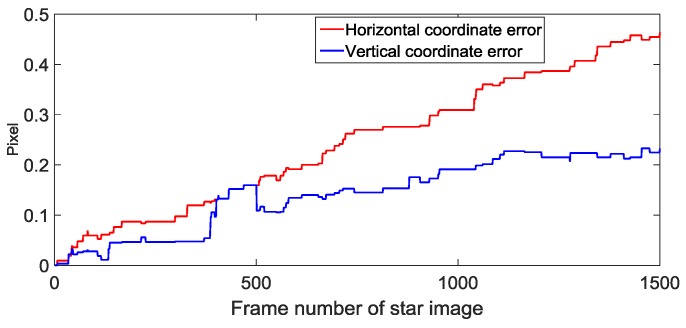
Predicted star position error.

**Figure 13 sensors-19-01890-f013:**
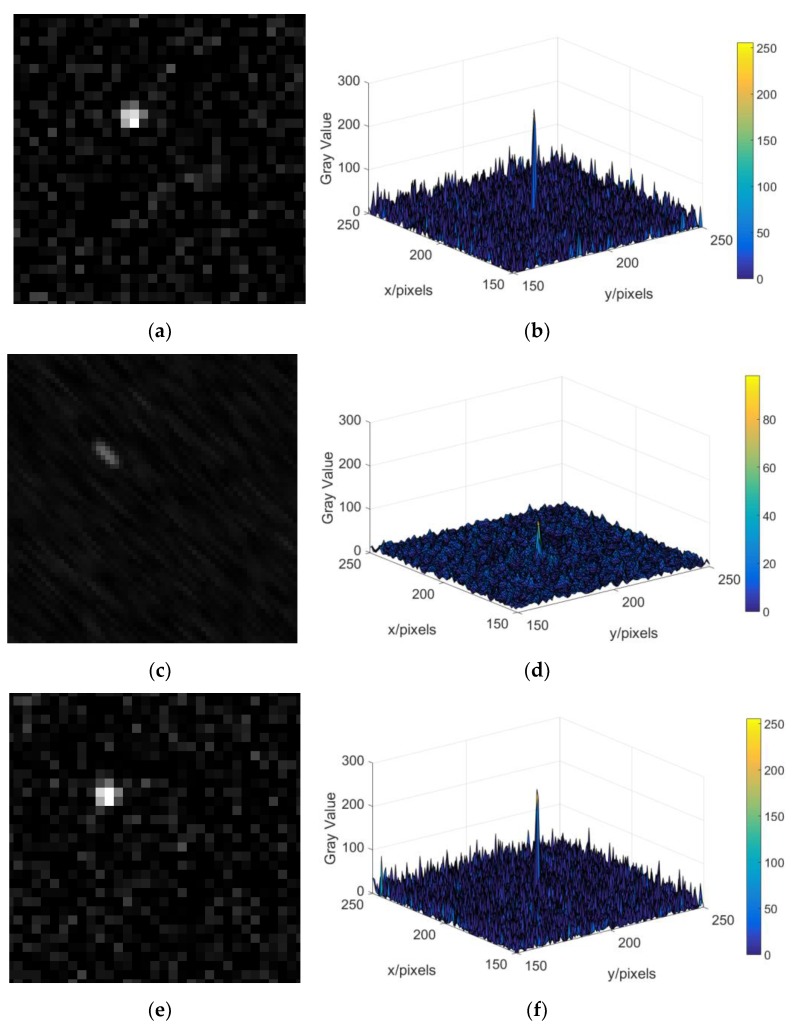
The magnified star image and the gray distribution of the star spot in the case of Gaussian white noise. (**a**) The magnified original star image; (**b**) gray value distribution of star spot in the original star image; (**c**) the magnified blurred star image ($w_{x}=w_{y}=10{}^{\circ}/s$); (**d**) gray value distribution of star spot in the blurred star image ($w_{x}=w_{y}=10{}^{\circ}/s$); (**e**) the magnified deblurred star image ($w_{x}=w_{y}=10{}^{\circ}/s$); (**f**) gray value distribution of star spot in the deblurred star image ($w_{x}=w_{y}=10{}^{\circ}/s$).

**Figure 14 sensors-19-01890-f014:**
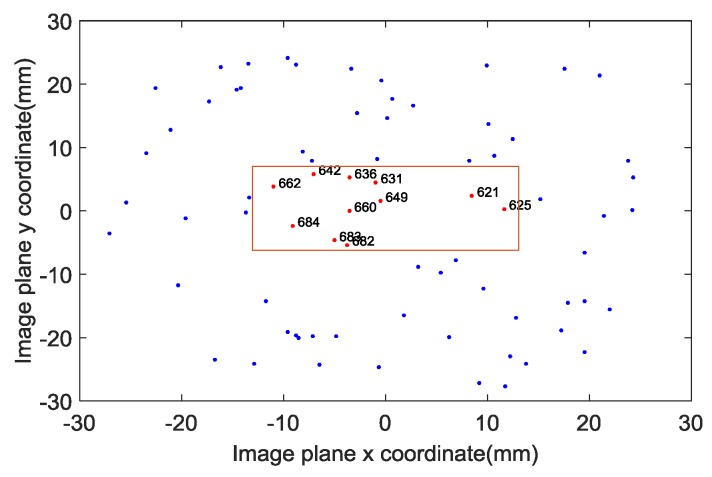
Star spots observed by a star sensor.

**Figure 15 sensors-19-01890-f015:**
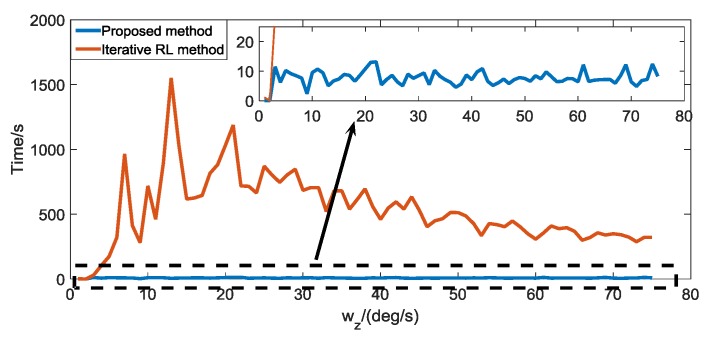
Comparison of running time between the proposed method and the iterative RL method in the case of Gaussian noise.

**Figure 16 sensors-19-01890-f016:**
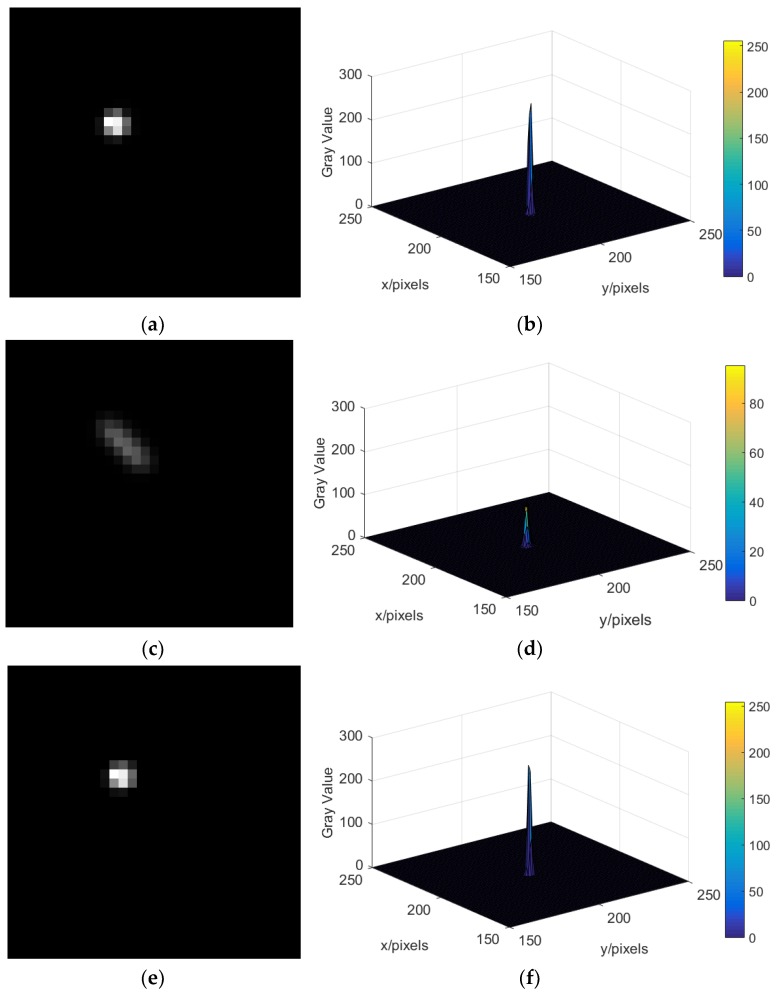
The magnified star image and the gray level distribution of the star spot in the case of Poisson noise. (**a**) The magnified original star image; (**b**) gray level distribution of star spot in the original star image; (**c**) the magnified blurred star image ($w_{x}=w_{y}=10{}^{\circ}/s$); (**d**) gray level distribution of star spot in the blurred star image ($w_{x}=w_{y}=10{}^{\circ}/s$); (**e**) the magnified deblurred star image ($w_{x}=w_{y}=10{}^{\circ}/s$); (**f**) gray level distribution of star spot in the deblurred star image (*w_x_* = *w_y_* = 10°/*s*).

**Figure 17 sensors-19-01890-f017:**
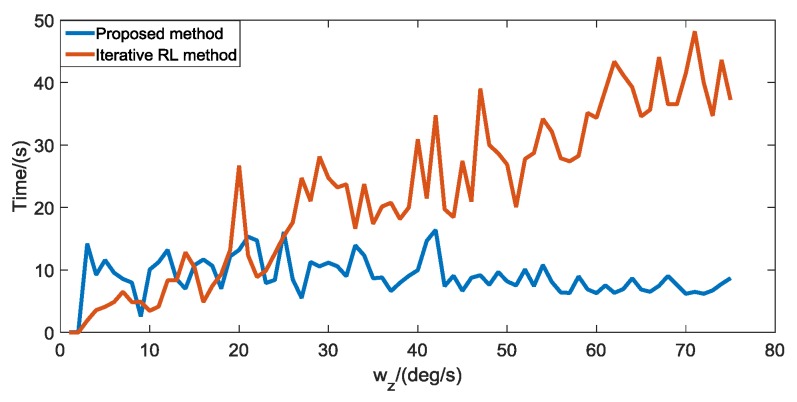
Comparison of running time between the proposed method and the iterative RL method in the case of Poisson noise.

**Table 1 sensors-19-01890-t001:** Comparison of the coordinates between the ideal and the predicted star spots in the 1500th star image.

Star Number	Ideal Star Spot Coordinate		Predicted Star Spot Coordinate		Predicted Star Spot Coordinate Error	
	x	y	$x^{\prime }$	$y^{\prime }$	Δx	Δy
1	24	63.50	23.25	63.62	0.75	−0.12
2	46.5	19	45.62	19.25	0.87	−0.25
3	54.50	371.50	54.23	371.76	0.26	−0.26
4	277	336.50	276.60	336.60	0.39	−0.10
5	290.50	305	290.11	305.27	0.38	−0.27
6	336.50	502.71	336.50	502.71	0	0
7	386.50	340	386	339.64	0.50	0.35
8	400.50	170	400	169.78	0.50	0.21
9	420.71	409.50	420.50	409.00	0.21	0.50
10	425.72	225.88	425.40	225.60	0.32	0.28
11	431.64	351	431.50	350.71	0.14	0.28
12	455	130.50	454.23	130.23	0.76	0.26
13	486.40	131.60	485.50	131.50	0.89	0.09

**Table 2 sensors-19-01890-t002:** Comparison of attitude estimation in the case of Gaussian white noise (Vary $w_{x}$).

$w_{x}\;(\deg/s)$	Attitude Errors (Blurred Star Image) (arc-second)			Attitude Errors (Restored Star Image by our Method) (arc-second)			Attitude Errors (Restored Star Image by Iterative RL Algorithm) (arc-second)		
	Pitch	Yaw	Roll	Pitch	Yaw	Roll	Pitch	Yaw	Roll
1	14.61	14.59	10.28	14.61	14.59	10.28	14.61	14.59	10.28
5	14.63	13.02	2.33	14.61	14.59	10.28	14.61	14.59	10.28
10	141.49	159.97	17.98	32.05	17.39	5.26	58.82	22.07	4.94
15	26.76	4.71	7.07	24.77	11.80	5.09	14.61	14.59	10.28
20	181.60	136.94	5.69	24.77	11.80	5.09	24.04	19.74	0.04
25	Fail	Fail	Fail	31.08	67.23	5.44	48.33	69.82	0.37
35	Fail	Fail	Fail	43.03	6.38	5.03	14.61	14.59	10.28
45	Fail	Fail	Fail	14.61	14.59	10.28	14.61	14.59	10.28
55	Fail	Fail	Fail	14.61	14.59	10.28	31.08	67.23	5.44
65	Fail	Fail	Fail	64.17	34.57	10.08	75.65	45.95	4.80
75	Fail	Fail	Fail	Fail	Fail	Fail	Fail	Fail	Fail

**Table 3 sensors-19-01890-t003:** Comparison of attitude estimation in the case of Gaussian white noise (Vary $w_{y}$).

$w_{y}\;(\deg/s)$	Attitude Errors (Blurred Star Image) (arc-second)			Attitude Errors (Restored Star Image by Our Method) (arc-second)			Attitude errors (Restored Star Image by Iterative RL Algorithm) (arc-second)		
	Pitch	Yaw	Roll	Pitch	Yaw	Roll	Pitch	Yaw	Roll
1	31.08	67.23	5.44	31.08	67.23	5.44	31.08	67.23	5.44
5	115.63	61.76	52.20	43.03	6.38	5.03	43.03	6.38	5.03
10	11.50	76.89	2.08	14.61	14.59	10.28	14.61	14.59	10.28
15	11.67	61.47	6.26	14.61	14.59	10.28	24.77	11.80	5.09
20	154.24	149.17	10.11	136.04	113.32	0.70	82.09	103.43	0.50
25	104.74	147.65	0.63	14.61	14.59	10.28	43.60	79.70	20.88
30	Fail	Fail	Fail	220.53	211.73	8.91	56.75	92.69	9.77
40	Fail	Fail	Fail	80.44	65.31	4.72	81.32	67.23	5.44
50	Fail	Fail	Fail	49.10	34.30	4.87	53.03	33.80	5.03
60	Fail	Fail	Fail	131.35	108.60	30.05	134.43	105.09	30.27
70	Fail	Fail	Fail	58.07	57.61	0.29	80.44	65.31	4.72
75	Fail	Fail	Fail	Fail	Fail	Fail	Fail	Fail	Fail

**Table 4 sensors-19-01890-t004:** Comparison of attitude estimation in the case of Gaussian white noise (Vary $w_{z}$).

$w_{z}\;(\deg/s)$	Attitude Errors (Blurred Star Image) (arc-second)			Attitude Errors (Restored Star Image by Our Method) (arc-second)			Attitude Errors (Restored Star Image by Iterative RL Algorithm) (arc-second)		
	Pitch	Yaw	Roll	Pitch	Yaw	Roll	Pitch	Yaw	Roll
1	14.61	14.59	10.28	14.61	14.59	10.28	14.61	14.59	10.28
5	103.24	73.51	12.20	4.38	40.73	5.28	14.61	14.59	10.28
10	60.43	55.49	25.06	14.61	14.59	10.28	14.61	14.59	10.28
15	157.89	162.40	15.28	14.61	14.59	10.28	14.61	14.59	10.28
20	84.80	136.06	3.85	14.61	14.59	10.28	14.61	14.59	10.28
25	Fail	Fail	Fail	14.61	14.59	10.28	14.61	14.59	10.28
35	Fail	Fail	Fail	14.61	14.59	10.28	34.94	19.01	19.09
45	Fail	Fail	Fail	30.03	66.27	20.87	14.61	14.59	10.28
55	Fail	Fail	Fail	14.61	14.59	10.28	91.86	76.68	4.65
65	Fail	Fail	Fail	75.67	96.89	0.44	112.50	111.72	16.04
75	Fail	Fail	Fail	92.57	106.52	5.68	170.85	140.59	4.22
80	Fail	Fail	Fail	Fail	Fail	Fail	Fail	Fail	Fail

**Table 5 sensors-19-01890-t005:** Comparison of attitude estimation in the case of Gaussian white noise (Vary $w_{x}$ and $w_{y}$).

Angular Velocity (deg/s)		Attitude Errors (Blurred Star Image) (arc-second)			Attitude Errors (Restored Star Image by Our Method) (arc-second)			Attitude Errors (Restored Star Image by Iterative RL Algorithm) (arc-second)		
$w_{x}$	$w_{y}$	Pitch	Yaw	Roll	Pitch	Yaw	Roll	Pitch	Yaw	Roll
1	1	31.30	28.17	16.23	14.61	14.59	10.28	14.61	14.59	10.28
5	5	203.95	191.12	22.83	13.14	30.54	0.11	14.61	14.59	10.28
10	10	29.66	18.84	4.72	14.61	14.59	10.28	14.61	14.59	10.28
15	15	335.46	369.20	31.02	14.61	14.59	10.28	14.61	14.59	10.28
20	20	Fail	Fail	Fail	14.61	14.59	10.28	14.61	14.59	10.28
30	30	Fail	Fail	Fail	58.82	22.07	4.94	56.75	92.69	9.77
40	40	Fail	Fail	Fail	14.61	14.59	10.28	4.68	39.04	15.47
50	50	Fail	Fail	Fail	49.10	34.30	4.87	41.74	9.47	10.16
60	60	Fail	Fail	Fail	14.61	14.59	10.28	31.08	67.23	5.44
70	70	Fail	Fail	Fail	126.33	125.57	0.77	120.24	97.62	0.61
75	75	Fail	Fail	Fail	Fail	Fail	Fail	Fail	Fail	Fail

**Table 6 sensors-19-01890-t006:** Comparison of attitude estimation in the case of Gaussian white noise (Vary $w_{x}$, $w_{y}$ and $w_{z}$).

Angular Velocity (deg/s)			Attitude Errors (Blurred Star Image) (arc-second)			Attitude Errors (Restored Star Image by Our Method) (arc-second)			Attitude Errors (Restored Star Image by Iterative RL Algorithm) (arc-second)		
$w_{x}$	$w_{y}$	$w_{z}$	Pitch	Yaw	Roll	Pitch	Yaw	Roll	Pitch	Yaw	Roll
1	1	1	66.08	27.77	16.17	14.61	14.59	10.28	14.61	14.59	10.28
5	5	5	56.02	46.97	37.18	14.61	14.59	10.28	14..61	14.59	10.28
10	10	10	101.42	84.97	12.21	58.82	22.07	4.94	31.08	67.23	5.44
15	15	15	Fail	Fail	Fail	5.46	41.82	20.65	14.61	14.59	10.28
25	25	25	Fail	Fail	Fail	83.89	119.70	9.66	42.49	49.42	25.86
35	35	35	Fail	Fail	Fail	80.51	43.55	20.27	176.13	2014.12	29.58
45	45	45	Fail	Fail	Fail	148.94	104.19	29.98	132.07	131.26	30.66
55	55	55	Fail	Fail	Fail	Fail	Fail	Fail	Fail	Fail	Fail

**Table 7 sensors-19-01890-t007:** Comparison of attitude estimation in the case of Poisson noise (Vary $w_{x}$).

$w_{x}\;(\deg/s)$	Attitude Errors (Blurred Star Image) (arc-second)			Attitude Errors (Restored Star Image by Our Method) (arc-second)			Attitude Errors (Restored Star Image by Iterative RL Algorithm) (arc-second)		
	Pitch	Yaw	Roll	Pitch	Yaw	Roll	Pitch	Yaw	Roll
1	14.61	14.59	10.28	14.61	14.59	10.28	14.61	14.59	10.28
20	32.73	32.90	15.31	14.61	14.59	10.28	14.61	14.59	10.28
35	47.34	86.91	27.38	14.61	14.59	10.28	15.05	16.10	14.61
40	Fail	Fail	Fail	14.61	14.59	10.28	14.61	14.59	10.28
55	Fail	Fail	Fail	108.82	137.30	5.82	65.28	94.19	26.05
70	Fail	Fail	Fail	81.34	73.45	5.61	2.64	12.00	20.58
85	Fail	Fail	Fail	5.77	37.94	0.10	33.45	76.93	15.70
100	Fail	Fail	Fail	14.61	14.59	10.28	24.77	11.80	5.09
115	Fail	Fail	Fail	32.61	11.32	30.77	58.32	14.25	15.18
130	Fail	Fail	Fail	77.30	40.39	20.17	155.73	162.04	11.18
155	Fail	Fail	Fail	43.03	6.38	5.03	115.90	122.36	10.87
160	Fail	Fail	Fail	Fail	Fail	Fail	Fail	Fail	Fail

**Table 8 sensors-19-01890-t008:** Comparison of attitude estimation in the case of Poisson noise (Vary $w_{y}$).

$w_{y}\;(\deg/s)$	Attitude Errors (Blurred Star Image) (arc-second)			Attitude Errors (Restored Star Image by Our Method) (arc-second)			Attitude Errors (Restored Star Image by Iterative RL Algorithm) (arc-second)		
	Pitch	Yaw	Roll	Pitch	Yaw	Roll	Pitch	Yaw	Roll
1	14.61	14.59	10.28	14.61	14.49	10.28	14.61	14.59	10.28
15	79.44	29.91	24.85	14.61	14.49	10.28	48.33	69.82	0.37
30	142.86	135.25	5.44	31.08	67.23	4.87	14.61	14.59	10.28
35	Fail	Fail	Fail	14.61	14.59	10.28	14.61	14.59	10.28
60	Fail	Fail	Fail	29.23	21.90	20.51	89.29	132.34	16.03
75	Fail	Fail	Fail	31.08	67.23	5.44	35.21	85.90	26.01
90	Fail	Fail	Fail	14.61	14.59	10.28	58.07	57.61	0.29
105	Fail	Fail	Fail	134.41	104.39	24.90	154.70	146.49	6.03
120	Fail	Fail	Fail	136.04	113.32	0.70	82.09	103.43	0.50
135	Fail	Fail	Fail	102.04	79.48	0.54	127.91	105.30	14.66
155	Fail	Fail	Fail	119.10	74.69	0.48	137.62	93.05	14.73
160	Fail	Fail	Fail	Fail	Fail	Fail	Fail	Fail	Fail

**Table 9 sensors-19-01890-t009:** Comparison of attitude estimation in the case of Poisson noise (Vary $w_{z}$).

*w_z_* (deg/s)	Attitude Errors (Blurred Star Image) (arc-second)			Attitude Errors (Restored Star Image by Our Method) (arc-second)			Attitude Errors (Restored Star Image by Iterative RL Algorithm) (arc-second)		
	Pitch	Yaw	Roll	Pitch	Yaw	Roll	Pitch	Yaw	Roll
1	14.61	14.59	10.28	14.61	14.59	10.28	14.61	14.59	10.28
15	124.07	82.00	14.15	14.61	14.59	10.28	15.01	14.94	0.09
30	171.49	129.07	26.77	14.61	14.59	10.28	14.61	14.59	10.28
35	Fail	Fail	Fail	14.61	14.59	10.28	14.61	14.59	10.28
45	Fail	Fail	Fail	14.61	14.59	10.28	4.38	40.73	5.28
60	Fail	Fail	Fail	14.61	14.59	10.28	49.10	34.30	4.87
75	Fail	Fail	Fail	14.52	58.08	20.74	28.99	7.05	15.25
90	Fail	Fail	Fail	101.47	129.93	4.50	115.88	64.16	9.72
105	Fail	Fail	Fail	22.90	22.86	25.87	88.02	109.29	15.91
120	Fail	Fail	Fail	31.08	67.23	5.44	33.47	54.99	0.21
150	Fail	Fail	Fail	15.34	15.26	0.09	14.61	14.59	10.28
165	Fail	Fail	Fail	87.48	43.19	14.98	75.17	74.57	9.85
170	Fail	Fail	Fail	Fail	Fail	Fail	Fail	Fail	Fail

**Table 10 sensors-19-01890-t010:** Comparison of attitude estimation in the case of Poisson noise (Vary $w_{x}$ and $w_{y}$).

Angular Velocity (deg/s)		Attitude Errors (Blurred Star Image) (arc-second)			Attitude Errors (Restored Star Image by Our Method) (arc-second)			Attitude Errors (Restored Star Image by Iterative RL Algorithm) (arc-second)		
$w_{x}$	$w_{y}$	Pitch	Yaw	Roll	Pitch	Yaw	Roll	Pitch	Yaw	Roll
1	1	40.61	18.59	22.13	14.61	14.59	10.28	14.61	14.59	10.28
15	15	51.70	33.10	21.44	14.61	14.59	10.28	14.61	14.59	10.28
25	25	372.32	260.78	13.32	14.61	14.59	10.28	49.10	34.30	4.87
30	30	Fail	Fail	Fail	14.61	14.59	10.28	43.08	6.38	5.03
45	45	Fail	Fail	Fail	14.71	14.46	10.27	14.71	14.46	10.27
60	60	Fail	Fail	Fail	43.60	79.70	20.88	43.03	6.38	5.03
75	75	Fail	Fail	Fail	80.44	65.31	4.72	75.75	118.93	16.02
90	90	Fail	Fail	Fail	52.75	59.83	25.98	58.82	22.07	4.94
105	105	Fail	Fail	Fail	58.82	22.07	4.94	60.44	25.31	4.72
115	115	Fail	Fail	Fail	138.30	159.19	0.82	138.30	159.19	0.82
120	120	Fail	Fail	Fail	Fail	Fail	Fail	Fail	Fail	Fail

**Table 11 sensors-19-01890-t011:** Comparison of attitude estimation in the case of Poisson noise (Vary $w_{x}$, $w_{y}$ and $w_{z}$).

Angular Velocity (deg/s)			Attitude Errors (Blurred Star Image) (arc-second)			Attitude Errors (Restored Star Image by Our Method) (arc-second)			Attitude Errors (Restored Star Image by Iterative RL Algorithm) (arc-second)		
$w_{x}$	$w_{y}$	$w_{z}$	Pitch	Yaw	Roll	Pitch	Yaw	Roll	Pitch	Yaw	Roll.
1	1	1	59.74	47.73	20.42	14.61	14.59	10.28	14.61	14.59	10.28
5	5	5	67.70	46.17	16.68	14.61	14.59	10.28	14..61	14.59	10.28
10	10	10	88.61	125.45	16.57	14.61	14.59	10.28	14.61	14.59	10.28
15	15	15	Fail	Fail	Fail	14.61	14.59	10.28	80.44	65.31	4.72
25	25	25	Fail	Fail	Fail	17.43	10.02	5.16	40.75	11.52	15.49
35	35	35	Fail	Fail	Fail	53.52	24.12	25.38	53.52	24.12	25.38
45	45	45	Fail	Fail	Fail	93.01	128.85	36.48	150.77	171.60	0.99
55	55	55	Fail	Fail	Fail	87.19	79.21	0.45	97.60	67.91	30.26
65	65	65	Fail	Fail	Fail	42.59	42.40	0.16	47.73	83.74	10.54
75	75	75	Fail	Fail	Fail	42.34	71.39	41.44	42.34	71.39	41.44
80	80	80	Fail	Fail	Fail	Fail	Fail	Fail	Fail	Fail	Fail

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
