# Peer review of "Star Image Prediction and Restoration under Dynamic Conditions"

_sensors, 2019, doi:10.3390/s19081890_

Round 1
Reviewer 1 Report
Thanks the authors for answering my questions. I have no further questions.
Please revise a typo I found, Line 306: EBPPNN.
Reviewer 2 Report
Authors have modified and included additional information that significantly improves the first submission and now it is acceptable for its publication.
This manuscript is a resubmission of an earlier submission. The following is a list of the peer review reports and author responses from that submission.
Round 1
Reviewer 1 Report
See attached file

Reviewer 2 Report
The authors proposed a parameter estimation approach for a star image restoration method, where they used neural networks to estimate the number of iterations for the RL algorithm. Here are my comments.
In page 10, please state reasons why they chose to use 1 x "n" weighting vector, where "n"=3. Did they try different "n"? Would it affect the results? Which n is better?
Just out of curiosity, in Fig. 11, could the authors try to explain why the vertical error are mostly smaller than the horizontal error?
I am not sure I understand what "Fail" in Tab. 2-6 means. Does that mean the error is too large that cannot be measured?
The authors mentioned about the iterative RL method, but only did a comparison in terms of running time. I suggest them to also test the iterative RL on attitude errors so that the readers could tell the performance difference between the iterative RL and the proposed method.
Reviewer 3 Report
This paper has several apparent drawbacks as follows:
1. The authors proposed a gyro-assisted star image prediction method to solve the frame loss problem of a star sensor. Now that the gyroscope is available, you can update the attitude of the spacecraft with gyros directly, which has been already widely used in many occasions. This method has no practical meaning and improves the computation load.
2. The RL algorithm is a relatively mature algorithm. The authors adopted EBPNN to predict the number of iterations. However, the simulations are not enough to affirmatively conclude that “the proposed methods are effective in improving the dynamic performance of the star sensor”. The authors should analyze whether the established model is still effective in other conditions(i.e. different background noises, dynamic conditions with varying angular velocity… ).
3. This paper lacks hardware-based experiments, which would be a good way to validate your conclusions.
4. “The gyroscope can be used for angular rates measurement at any angular velocity with advantages of small size, lightweight, and easy to integrate”. This assertion is not sound.
5. In Figure 2, the star sensor coordinate system doesn’t satisfy the right-hand rule.
6. The Eq.11 is the same as Eq.9.
7. In Eq.23, generally, the rotation matrix from the gyroscope coordinate system to the star sensor coordinate system is not an identity matrix. It can be calibrated in advance.
8. The field of view of the star tracker is determined by the focal length and the detector size. As given in Section 4.1, the detector size is 512*512, the pixel size is 20um, the focal length is 49mm, how can the field of view be 20°?
9. Generally, due to the difference between the detector size and the focal length, star smearing induced by rotations along boresight direction is more serious than that along the boresight direction. In your simulations, why the errors of blurred star image with varying wz (See Table 4) are even worse than those with varying wx and wy(See Tables 2 and 3).
10. The English should be improved.